# Performance of screening tools for cervical neoplasia among women in low- and middle-income countries: A systematic review and meta-analysis

**Sabrina K. Smith[1], Oguchi Nwosu[2], Alex Edwards[3], Meseret Zerihun[4], Michael H. Chung[1,5,6], Kara Suvada[5‡]\*, Mohammed K. Ali[1,2‡]**

1 Hubert Department of Global Health, Rollins School of Public Health, Emory University, Atlanta, Georgia, United States of America, 2 Department of Family and Preventive Medicine, School of Medicine, Emory University, Atlanta, Georgia, United States of America, 3 Department of Biostatistics and Bioinformatics, Rollins School of Public Health, Emory University, Atlanta, Georgia, United States of America, 4 Department of Family Medicine, Addis Abba University, Addis Abba, Ethiopia, 5 Department of Epidemiology, Rollins School of Public Health, Emory University, Atlanta, Georgia, United States of America, 6 Division of Infectious Diseases, Department of Medicine, Emory University, Atlanta, Georgia, United States of America

‡ KS and MKA share co-senior authors on this work.
\* kara.suvada@emory.edu

**Data Availability Statement:** All data are available in the Supplemental Information files.

## Abstract

### Objective

To evaluate the performance of visual inspection with acetic acid (VIA) testing, visual inspection with Lugol's iodine (VILI), primary HPV testing, and conventional Pap smear in detecting CIN2+ among non-pregnant women aged 30–65 in LMICs between 1990 and 2020.

### Design

Systematic review and meta-analysis.

### Setting and participants

Low- and middle-income countries, non-pregnant women aged 30–65.

### Methods

CENTRAL (Cochrane Library), CINAHL, Embase, Global Health, PubMed, and Web of Science databases were systematically searched to identify studies evaluating the performance of cervical cancer screening methods in LMICs. A diagnostic test accuracy meta-analysis was conducted to evaluate the performance of 4 screening methods in detecting CIN2+ relative to biopsy or cytology reference standards. Pooled statistics for sensitivity, specificity, diagnostic odds ratios, and summary receiver operating characteristic curves were determined for each method. Subgroup analyses were performed to examine whether there was variation in performance based on different reference standards for defining

**Funding:** The authors received no specific funding for this work.

**Competing interests:** The authors have declared that no competing interests exist.

CIN2+, specifically: colposcopy-directed biopsy, biopsy alone, colposcopy alone, or liquid-based cytology.

## Results

Eighteen studies were identified through systematic review. Twelve studies were included in meta-analysis; 11 were cross-sectional and 1 was a randomized controlled clinical trial. The remaining six of the eighteen studies were inclided in a narrative syntehsis. Pooled estimates for sensitivity for VIA, VILI, primary HPV testing, and conventional Pap smear were 72.3%, 64.5%, 79.5%, and 60.2%, respectively; pooled estimates for specificity were 74.5%, 68.5%, 72.6%, and 97.4%, respectively; the diagnostic odds ratios were 7.31, 3.73, 10.42, 69.48, respectively; and the area under the summary receiver operating characteristic curves were 0.766, 0.647, 0.959, and 0.818, respectively. Performance of the screening method varied based on the reference standard used; pooled estimates using either colposcopy-directed biopsy or biopsy alone as the reference standard generally reported lower estimates; pooled estimates using either colposcopy alone or liquid-based cytology as references reported higher estimates.

## Conclusions and implications

This meta-analysis found primary HPV testing to be the highest performing cervical cancer screening method in accurately identifying or excluding CIN2+. Further evaluation of performance at different CIN thresholds is warranted.

## Introduction

Cervical cancer is preventable; despite this, on a global scale, it is the fourth most common cancer in women and leads all gynecologic cancers in mortality [1]. Most cases and deaths occur in low- and middle-income countries (LMICs) [2–6]. In 2020, approximately 604,000 women received a cervical cancer diagnosis, and an estimated 342,000 women died from the disease [1]. Following the introduction of organized cervical cancer screening in the 1960s, high-income countries have experienced a steady decline in cervical cancer rates [7]. The age-standardized incidence rate (ASIR) for cervical cancer in resource-rich countries is 10/100,000 compared to 25 to 55/100,000 in resource-limited countries [8]. In the United States, cervical cancer incidence and mortality have decreased by more than 70% since the 1950s and the decline in cervical cancer death rates is largely credited to screening programs [9, 10]. However, the burden of cervical cancer remains prevalent in many LMICs where screening programs are either unavailable or poorly implemented. According to the World Health Survey, the mean crude coverage of cervical cancer screening in LMICs was reported to be 45%, and effective coverage (a measure that combines intervention need, use, and quality) was reported at 19% [11].

To increase screening coverage in LMICs, it is essential to identify and implement feasible and cost-effective screening strategies appropriate for these settings. Existing methods include: unaided visual inspection (UVI), visual inspection with acetic acid (VIA), or visual inspection with Lugol's iodine (VILI). The most common practice in LMIC settings is VIA which is considered cost-effective and feasible–requiring less provider training and produces immediate results, reducing patient loss to follow-up and allowing for a screen-and-treat

approach [12]. That said, there are limited data comparing the performance of VIA with more commonly used screening methods such as the Pap smear with or without reflex HPV testing and primary HPV testing (either laboratory-based or point-of-care) alone [13]. VIA or VILI alone or rapid resulting point-of-care HPV testing or a combination of these have the potential to transform cervical cancer screening to a screen-and-treat approach in resource limited settings. Here, we systematically reviewed the literature to compare the average performance of different screening tools to detect cervical intraepithelial neoplasia grade 2 and above in LMIC settings.

## Materials & methods

### Study search and selection

The CENTRAL (Cochrane Library), CINAHL, Embase, Global Health, PubMed, and Web of Science databases were systematically searched to identify studies assessing test performance of VIA, VILI, primary HPV testing, and Pap smear, published from January 1, 1990 to December 31, 2020. These databases were searched using Medical Subject Heading (MeSH) terms such as "cervical cancer", "visual inspection", "pap smear", "developing country", "mass screening", and "clinical outcome". Publications were restricted to the English language. The review protocol was registered in PROSPERO (CRD42020206154) and the report adheres to PRISMA (Preferred Reporting Items for Systematic Reviews and Meta-Analyses) guidelines for reporting systematic reviews [14]. A review protocol was not prepared.

The study population of interest included non-pregnant women aged 30–65 in low- and middle-income countries (LMICs) who received VIA testing and either VILI, Pap smear, or primary HPV test between 1990–2020. The decision to exclude studies was because women under 30 have a low prevalence of underlying high-grade lesions and a high prevalence of transient HPV infection, meaning that they contract and eradicate HPV more quickly than women 30 years and older [15, 16]. Studies that included participants who were pregnant or had a history of hysterectomy were excluded from this review. Studies with an inclusion criterion specifying that participants be symptomatic were also excluded from this review as those patients would require diagnostic testing and not screening. Additionally, studies that utilized co-testing, as opposed to primary HPV testing, were excluded. HPV co-testing includes both a Pap test and HPV test performed at the same time, while primary HPV tests involve a singular HPV test. Only primary HPV test results were collected in order to isolate the test results and determine screening performance of the singular test.

This review evaluated diagnostic accuracy at the CIN2+ (cervical intraepithelial neoplasia grades 2 and higher), given the higher likelihood of CIN2+ progression to cervical cancer [17]. Studies included in this review consisted of randomized controlled clinical trials, cross-sectional studies, and cohort studies. Systematic reviews and meta-analyses were excluded from this review due to differing inclusion criteria and to avoid duplication of studies included in this review. Good and fair quality studies were assessed for inclusion in this review, as defined by the National Institute of Health (NIH) Quality Assessment Tool [18]. Data that was available only in abstract form or grey literature were not eligible.

Covidence software was used for data management throughout this review [19]. Two reviewers independently screened all study titles and abstracts identified through MeSH database searches and selected studies for full-text review. Reviewers then individually screened all full texts to select studies for data extraction. Disagreements between independent reviewers were adjudicated by a third reviewer for title and abstract screening, full-text review, and data extraction.

## Data extraction

A standardized data extraction sheet was designed to extract data relevant to the review. Data extraction was primarily conducted by author SS and validated by authors KS and MA using a decision tree approach. In addition, author KS extracted a subsample of 10% of studies to check for variation in extraction. The primary outcome included extraction of the sensitivity and specificity of VIA, VILI, primary HPV tests, and Pap tests; VILI was the CIN2+ threshold. When raw data was available, including true positives, true negatives, false positives, and false negatives, these results were extracted. When only computed data for sensitivity and specificity was available, individual raw data were calculated based on identified proportions. When studies reported CIN2 and CIN3 as separate thresholds, these measures were weighted and combined to determine sensitivity and specificity of the combined CIN2+ threshold. Measures that were reported as HSIL on the Bethesda scale were converted to the CIN reporting method and denoted as CIN2+ [17]. When studies reported outcomes for aggregates by age group, outcomes were either extracted or backed into using raw true positive (TP), true negative (TN), false positive (FP), false negative (FN) for the desired age group ($\geq$ 30 years). Secondary outcomes included the extraction of positive predictive value, negative predictive value, adverse effects, and likelihood ratios. Information was also extracted regarding the study-level characteristics and participant-level characteristics.

## Data synthesis and analyses

This study utilizes the diagnostic test accuracy (DTA) approach to quantitatively synthesize extracted data [20, 21]. Through DTA, representative pooled statistics of sensitivity and specificity are combined into one effect size. Additional representative pooled statistics generated include the diagnostic odds ratio (DOR) and forest plot, as well as the pooled receiver operating characteristic (SROC) curve. The DOR is the odds of a positive diagnostic test among those who actually have cervical cancer relative to the odds of a positive diagnostic test among those who do not have cervical cancer. The SROC is a curve plot with the sensitivity as the y-axis and the false positive rate (i.e., one minus the values of the specificity) as the x-axis. These representative pooled statistics and pooled lines are derived from raw data including the true positive, false positive, false negative, and true negative values that make up a 2 x 2 table [21].

To perform the DTA approach, raw data (TP, FP, FN, TN) was coded in RStudio (version 1.2.5042) to generate pooled statistics of sensitivity, specificity and DOR, as well as a summary line (SROC curve) [21, 22]. The R "mada" package reitsma model was used to calculate these pooled statistics using the bivariate model, estimating pooled measures of sensitivity and specificity separately for each comparison group while accounting for the potential correlation between sensitivity and specificity [23]. The bivariate model is similar to the random effects meta-analysis model of a pair-wise comparison, and is able to estimate heterogeneity, or the within-study and between-study variation of studies. This model assumes a binomial distribution to model representative pooled statistics for within-study variation, and a bivariate normal distribution for between-study variation [21]. The bivariate approach produces unbiased estimates of sensitivity, specificity, and their correlation, and does not rely on ad hoc continuity correction for zero marginal counts [22]. A subgroup analysis was also performed, generating pooled estimates for each comparison group by reference standard. Summary receiver-operating characteristics curves (SROC) were obtained along with 95% confidence regions for the bivariate estimates of sensitivity and 1-specificity. This curve indicates how discriminating a model is, and how well one is able to discern an individual study from another study included in the review [20].

Following the generation of pooled statistics, heterogeneity was verified and reported. Heterogeneity could be due to chance, difference in cut-off value, difference in study design, prevalence, research environment, and demographic factors of the sample population [20]. To assess statistical heterogeneity, the Higgins' $I^2$ measure was quantified, indicating the percentage of total variation across studies due to heterogeneity rather than chance [24]. Additionally, the symmetry and scattering of the SROC curve was assessed, and the correlation coefficient of sensitivity and specificity was calculated using the R "mada" package [25, 26].

The R "meta" package was used to calculate the total effect sizes of pooled statistics through univariate analysis, such as the combined sensitivity for all studies included in the VIA comparison group [27]. This data was then plotted via "mada" on an SROC curve utilizing bivariate analysis [23]. This was a necessary step given the limitations of the "mada" package in calculating total effect sizes and allowed for each pooled statistic value to be verified through univariate analysis in addition to bivariate analysis. The R "meta" and "mada" packages have been described in depth at https://cran.r-project.org/web/packages/meta/meta.pdf and https://CRAN.R-project.org/package=mada [21, 28].

## Results

### Characteristics of all included studies

This search strategy generated 1,518 citations, with 1 additional study added for consideration for inclusion in the review. Following abstract screening and full-text review, 18 studies met eligibility criteria and 12 studies included sufficient data to be pooled for meta-analysis [29–40]. Fig 1 shows the flowchart of study selection according to PRISMA [14] (Fig 1 and S1 Fig). The 6 studies excluded from meta-analysis but included in narrative synthesis fit the inclusion criteria and scope of the review but did not explicitly state the raw data needed to conduct meta-analysis, including true positive, false positive, true negative, and false negative values per group. These studies were narratively synthesized to provide relevant context to the review [41–46].

The N = 12 studies included in meta-analysis were published between 2004 and 2020. The pooled sample size from these studies totaled N = 110,657 participants and ranged from n = 100 to n = 54,981 women per study. Participant age ranged from 18 years to 65 years, with all studies reporting either a mean or median ≥ 35 years (Table 1). These studies were geographically diverse, representing India (1; 8.3%), Zambia (1; 8.3%), Kenya (1; 8.3%), Papua New Guinea (1; 8.3%), China (3; 25.0%), Iran (1; 8.3%), Bangladesh (1; 8.3%), Mongolia (1; 8.3%), and the Democratic Republic of the Congo (1; 8.3%). One study was multinational (1; 8.3%) and represented participants from Mali, the Congo Republic, Guinea, Niger, Burkina Faso, and India (Table 1 and S1 to S4 Data). The quality of all studies was rated as high, as determined by the NIH Quality Assessment Tool (S1 Table) [13]. Eleven (91.7%) studies were cross sectional and 1 (8.33%) was a randomized controlled trial.

Studies employed 4 different screening approaches to detect pre-cancerous lesions of the cervix at the CIN2+ threshold: VIA, VILI, primary HPV testing, and Pap smear (Table 2). All of the studies included in this review conducted VIA as a screening method; 3 (25%) of the studies conducted VILI; 5 (41.7%) of the studies conducted primary HPV testing; and 5 (41.7%) of the studies conducted Pap smears.

Studies used a variety of reference standards to determine the diagnostic accuracy of each of these methods. Reference standards included colposcopy-directed biopsy (50% of studies), biopsy alone (33.3% of studies), colposcopy alone (8.3% of studies), and liquid-based cytology (LBC) (8.3% of studies). Test characteristics for the CIN2+ threshold were aggregated based

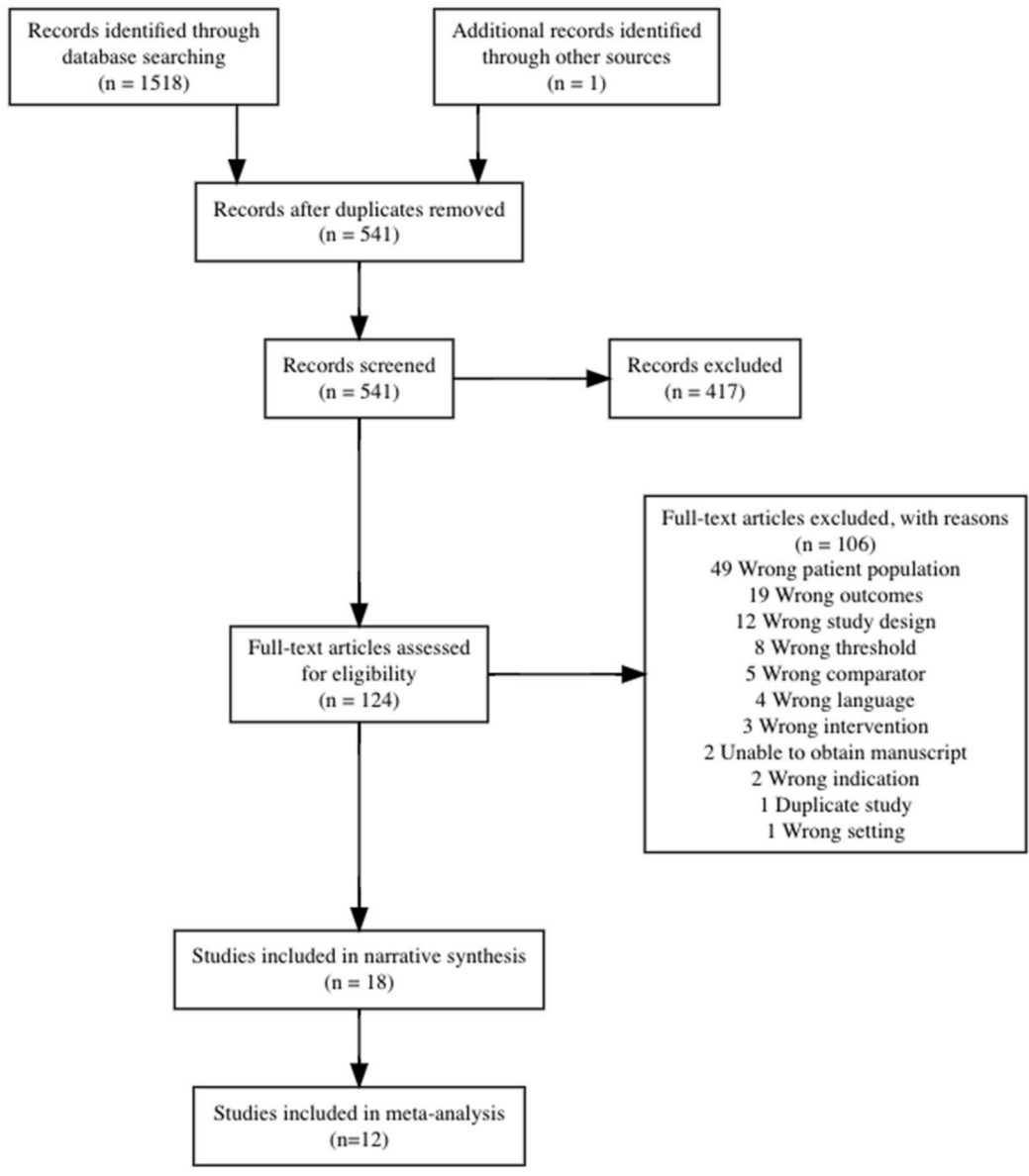

**Fig 1. PRISMA diagram.**

on screening method and reference group, as displayed in Figs 2–5. All studies assessed the sensitivity and specificity of one or more groups for detecting CIN2+.

## Subgroup analysis: VIA comparison group

The VIA screening approach had a combined sensitivity of 0.723 [0.641; 0.792], with lower sensitivity noted when the reference standard was LBC. Specificity of VIA was, on average, 0.745 [0.569; 0.866], and did not vary significantly by reference standard. The studies reporting on VIA were generally heterogenous. The combined DOR for the VIA comparison group was 7.3078 [3.6547; 14.6122], with a higher DOR reported when the reference group was colposcopy (37.5711 [15.4299; 91.4842]) (Fig 2).

**Table 1. Pooled characteristics of participants in included studies (N = 12 studies and N = 110,657 participants).**

| Characteristic | Measurement |
|---|---|
| **Age** | Median: > 34 years |
| | Mean: > 34 years |
| | Range: 18–65 years |
| **Participants per Study** | Range: 100–54,981 |
| **Country** | |
| Bangladesh | 1/12 (8.3%) |
| China | 3/12 (25.0%) |
| Democratic Republic of the Congo | 1/12 (8.3%) |
| India | 1/12 (8.3%) |
| Iran | 1/12 (8.3%) |
| Mongolia | 1/12 (8.3%) |
| Multinational (Burkina Faso, Congo Republic, Guinea, Mali, Niger, India) | 1/12 (8.3%) |
| Papua New Guinea | 1/12 (8.3%) |
| Zambia | 1/12 (8.3%) |
| **Date Range** | 2004–2020 |
| **Type of Study** | |
| Cross-Sectional | 11/12 (91.7%) |
| Randomized Control Trial | 1/12 (8.33%) |

## Subgroup analysis: VILI comparison group

From 2 studies, the VILI screening approach had a combined sensitivity of 0.645 [0.571; 0.713] and did not vary by reference standard. Specificity of VILI was, on average, 0.685 [0.460; 0.847]; reported at 0.563 [0.391; 0.722] and 0.856 [0.846; 0.865] for colposcopy-directed biopsy and biopsy, respectively. The studies reporting on VILI were somewhat heterogenous. The combined DOR for the VILI comparison group was 3.7331 [0.8797; 15.8418], with a higher DOR reported when the reference group was biopsy (10.7737 [7.4001; 15.6853]) (Fig 3).

## Subgroup analysis: Primary HPV test comparison group

In 5 studies assessing primary HPV screening, sensitivity was 0.795 [0.604; 0.908] and a higher sensitivity observed when the reference standard was LBC. The combined specificity of primary HPV testing was 0.726 [0.340; 0.932] but varied substantially for the colposcopy-directed biopsy reference group, with a specificity of 0.376 [0.213; 0.572]. The studies reporting on primary HPV testing were generally heterogenous. The combined DOR for the primary HPV test comparison group was 10.4183 [1.7443; 62.2257], with a higher DOR reported when the reference group was colposcopy (101.9792 [30.1424; 345.0202]) (Fig 4).

**Table 2. Descriptions of screening approaches.**

| Name of Screening Approach | Description |
|---|---|
| Visual Inspection with Acetic Acid (VIA) | Provider swabs patient's cervix with acetic acid solution (e.g., vinegar); in response, pre-cancerous lesions will turn white and allow for direct visual inspection [4] |
| Visual Inspection with Lugol's Iodine (VILI) | Provider swabs patient's cervix with Lugol's iodine; using the Lugol's iodine and a light source, a provider can visually distinguish between precancerous lesions and healthy tissue [4]. |
| Primary HPV testing | Cervical cells are collected and sent to a lab for testing for HPV [47]. |
| Pap smear | Cervical cells are collected and sent to a lab for testing for abnormal growth [47]. |

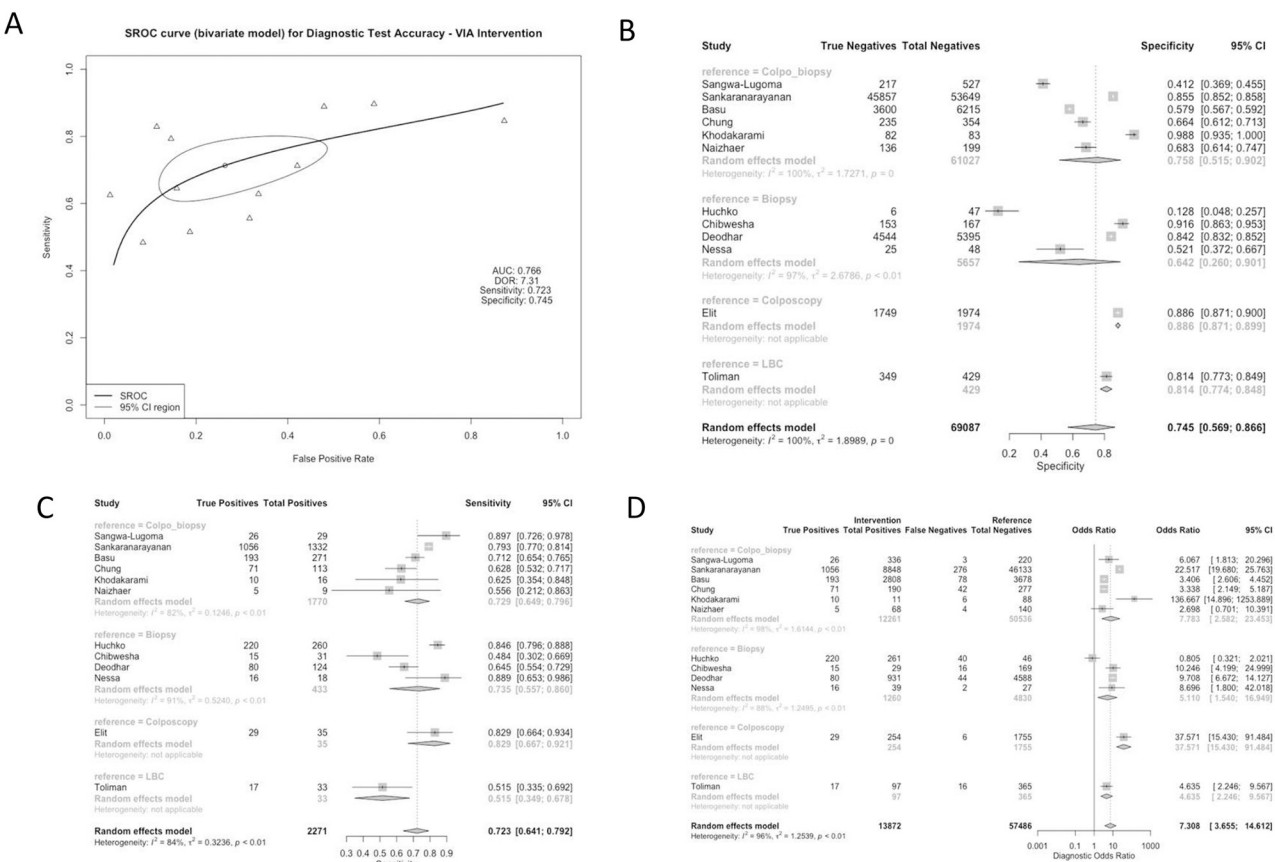

**Fig 2.** Panel A: Pooled Receiver Operating Characteristic for Diagnostic Test Accuracy for VIA intervention. Panel B: Specificity of Studies for VIA intervention. Panel C: Sensitivity of Studies for VIA intervention. Panel D: Diagnostic Odds Ratio for VIA intervention.

## Subgroup analysis: Pap smear comparison group

From a total of 5 studies, the Pap smear screening approach had a combined sensitivity of 0.602 [0.361; 0.803], with higher sensitivity noted when the reference standard was colposcopy alone. Specificity of Pap smear was, on average, 0.974 [0.955; 0.985], and did not vary by reference standard. The studies reporting on Pap smear were mostly heterogenous. The combined DOR for the Pap smear comparison group was 69.4863 [25.6440; 188.2837], with a higher DOR reported when the reference group was colposcopy (499.4741 [165.5978; 1506.5078]) (Fig 5).

## Narrative synthesis

Six studies fit the inclusion and exclusion criteria for the systematic review but were not included in meta-analysis because of limitations in data reported in the respective manuscripts. These 6 studies included data from India, China, Iran, Brazil, and Argentina [22, 24, 28, 41, 42] Among these studies, sensitivities for VIA ranged from 43% to 94.6%; specificity for VIA demonstrated less variation, ranging from 81.6% to 96.7%. These variations in diagnostic accuracy of VIA reflect the considerable heterogeneity observed in the meta-analysis of findings.

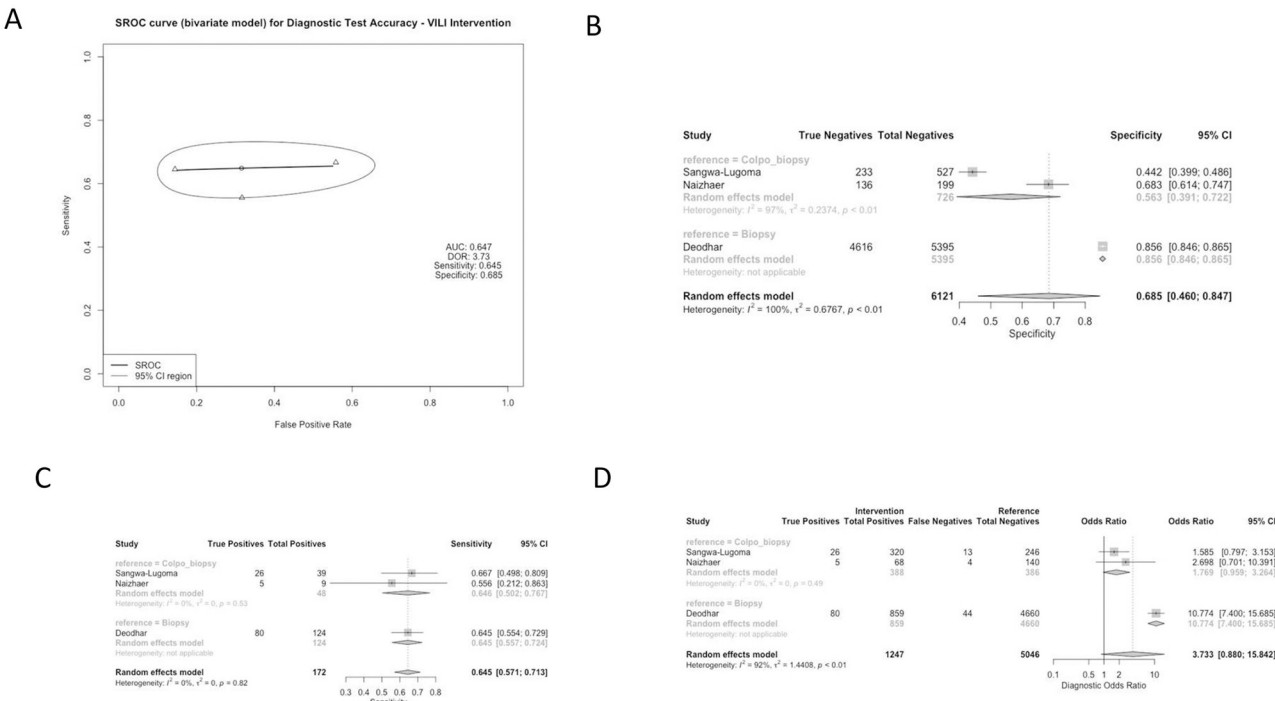

**Fig 3.** Panel A: Pooled Receiver Operating Characteristic for Diagnostic Test Accuracy for VILI intervention. Panel B: Specificity of Studies for VILI intervention. Panel C: Sensitivity of Studies for VILI intervention. Panel D: Diagnostic Odds Ratio for VILI intervention.

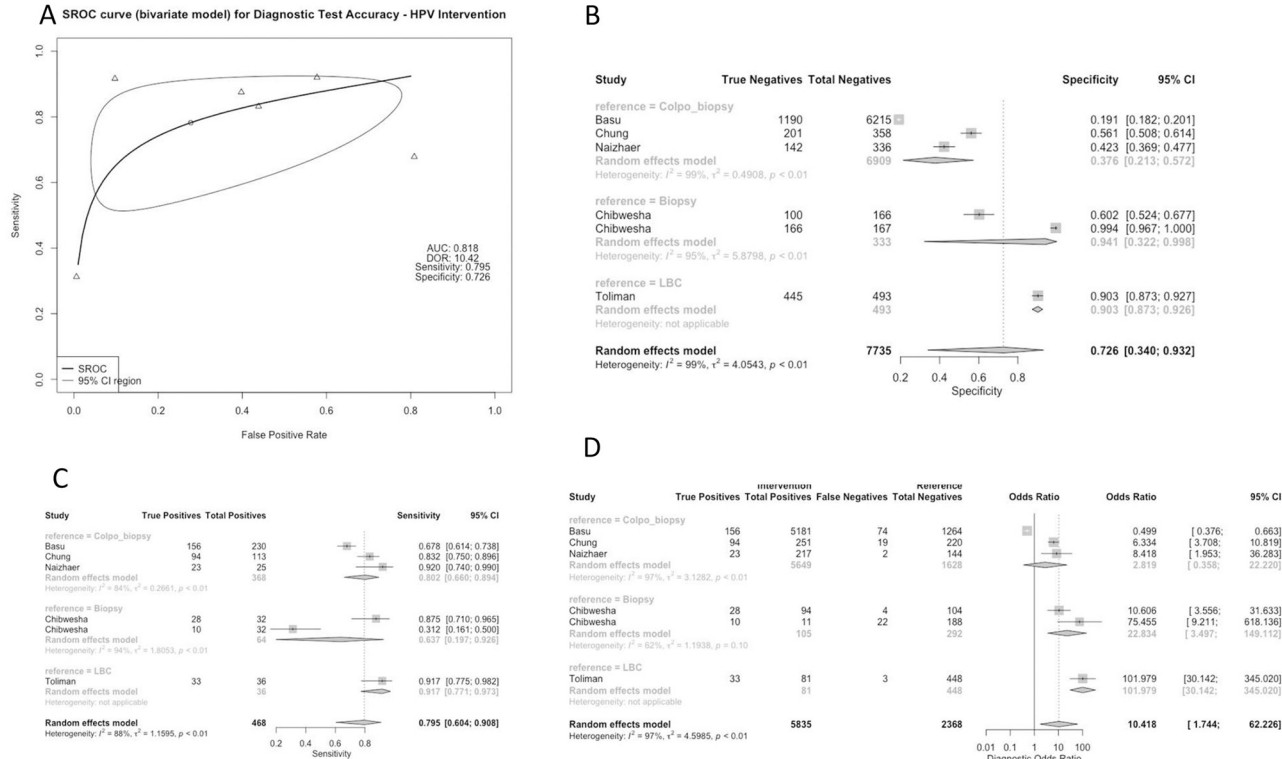

**Fig 4.** Panel A: Pooled Receiver Operating Characteristic for Diagnostic Test Accuracy for HPV Intervention. Panel B: Specificity of Studies for HPV Intervention. Panel C: Sensitivity of Studies for HPV Intervention. Panel D: Diagnostic Odds Ratio for HPV Intervention.

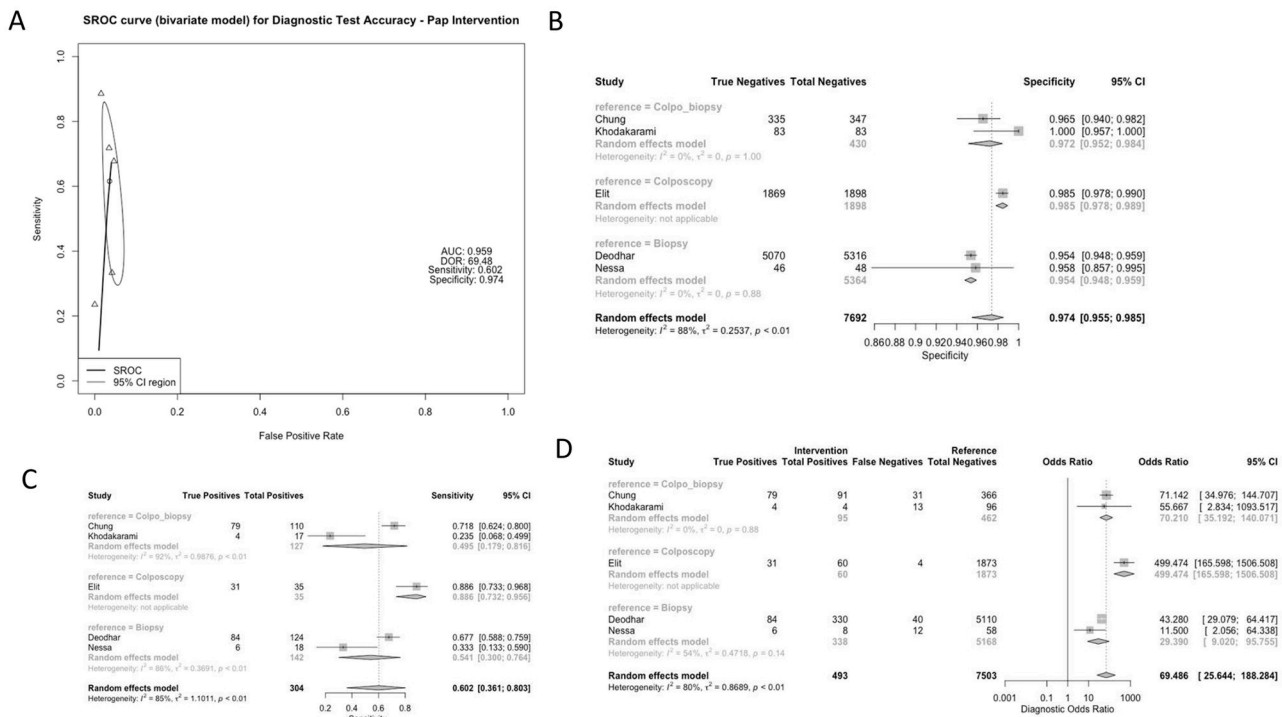

**Fig 5.** Panel A: Pooled Receiver Operating Characteristic for Diagnostic Test Accuracy for Pap smear intervention. Panel B: Specificity of Studies for Pap smear intervention. Panel C: Sensitivity of Studies for Pap smear intervention. Panel D: Diagnostic Odds Ratio for Pap smear intervention.

## Discussion

To our knowledge, this was the first systematic review and meta-analysis evaluating the diagnostic performance of cervical cancer screening tests in non-pregnant women living in LMICs. Disease status was confirmed with either colposcopy-directed biopsy, biopsy alone, colposcopy alone, or liquid-based cytology. Results indicated that VIA and primary HPV testing had similar sensitivity, specificity, and diagnostic odds ratios; pap smears offered higher specificity, but lower sensitivity. VILI had the worst diagnostic performance, appearing to be the most undesirable screening method of the four methods evaluated. These findings have major implications for screen-and-treat guidelines in LMICs that need to occur in parallel with HPV vaccination to address global cervical cancer incidence and mortality [4].

Conventional Pap smear continues to be the most widely used cervical cancer screening method worldwide, despite constraints with the sensitivity of the test. This study demonstrated low sensitivity which aligns with other meta-analyses that have evaluated performance and test characteristics of Pap smear [1, 43, 44]. The low sensitivity associated with Pap smear appears to be attributable to the nature of the test, variation in how the procedure is done or the slide is made or the histology read, and necessitates the use of co-testing with HPV testing or rescreening for cervical cancer annually or bi-annually. Additionally, pap smears do not allow for a screen-and-treat approach, requiring patient follow-up after the initial appointment which proves to be a barrier for patients in LMIC settings where travel and transport may be an issue [37, 45].

The highest performing diagnostic test in this meta-analysis was primary HPV testing. However, for laboratory-based primary HPV testing, cost-effectiveness, loss to follow-up, and inadequate testing conditions continue to be barriers in LMIC settings [21, 46]. If rapid

resulting point-of-care HPV testing were more widely available and performed similarly, it is likely that this would be the test of choice. Point-of-care primary HPV test kits allow for both provider-collected samples as well as self-sampling in a health care setting or at home, increasing convenience and patient comfort. Furthermore, the ability to make a diagnosis during one visit provides the opportunity for a 'screen and treat' protocol using methods like cryotherapy or thermocoagulation for treatment.

While all methods are relatively safe, the added discomfort during VIA and pap procedures may discourage women, particularly in settings where follow-up services are less accessible. It is also important to note the burden of misdiagnosis and over-screening. False-negative reports, or failing to identify CIN, contributes to increased rates of cervical cancer [48]. Alternatively, false-positive reports, or inaccurately characterizing cells as abnormal, are shown to cause both physical and psychological burden in patients, including anxiety and unnecessary invasive investigative procedures and treatment [9]. In both cases, inaccurate results contribute to patient harm and over-screening, leading to increased cost and patient mistrust in screening practices [9]. An idyllic cervical cancer screening test would be able to be performed in a primary healthcare facility, conducted with limited training and technology, and produce rapid (e.g., same-day, point-of-care) results. These outcomes stress the importance of a high-performing screening test to minimize harm in patient populations [43].

With the exception of the VILI group, results indicated considerable heterogeneity for VIA, Pap, and primary HPV screening when analyzing SROC curves. This finding suggests that there is very little similarity between studies in each comparison group. Studies were all high quality as determined by the NIH Quality Assessment Tool, and comparisons were performed in similar patient populations with a mean $\geq$ 35 years of age living in LMICs [18]. Heterogeneity may be a function of study designs, the dataset, the available data, or a combination of these issues. Further analysis should be done to confirm whether certain variables contribute to increased heterogeneity in outcomes, such as the study region, study size of the population, or capacity of test providers [28].

One of the major strengths of this systematic review and meta-analysis is the ability to examine test characteristics in the population of patients aged 30–65 years. Focusing on this patient population addresses CIN findings that are more likely to progress to cervical cancer and less likely to resolve without treatment [15]. The scope of this review allowed for any LMIC to be included, contributing to the breadth of the dataset and evaluating performance of these screening tests on a global stage. This review helped to identify gaps in the literature; the majority of studies evaluated screening methods in peri-urban and urban areas of middle-income countries. Future studies should explore test performance of screening methods in expanded geographies, to include rural populations, low-income countries, and high-income countries. Additionally, most studies did not report outcomes beyond sensitivity, specificity, positive predictive value, and negative predictive value. Studies should expand reporting to include values such as likelihood ratios and DOR to aid in the assessment of clinical utility of the results.

This study has several limitations that should be considered when interpreting these results. By evaluating performance of test characteristics solely at the CIN2+ threshold, performance of screening methods at ASCUS and CIN1 thresholds were missed. Evaluating performance at multiple thresholds would help to provide context for results at the CIN2+ threshold, as well as contribute to the body of evidence supporting the use of certain screening methods. Another limitation related to threshold was the standardization of the Bethesda system under the CIN reporting system umbrella. While this crosswalk allowed for standardized comparisons groups, studies using the Bethesda system to report CIN may have had slight variations in CIN diagnosis by clinicians and using various methods to establish the reference group could not be accounted for. The studies included in this review did not always stratify data by pregnancy

status as outlined in the exclusion criteria, including data from the LAMS study in the narrative synthesis, which had a 3.5% pregnant population at the time of the screening [49, 50]. The studies included in this meta-analysis and literature review also did not consistently report HPV serotypes. We also did not stratify by HIV status in our analyses because not all studies reported HIV status of women; however, a recent meta-analysis was conducted that examined similar outcomes to our study among only women living with HIV [51]. Additionally, manual calculations and rounding of estimates in the literature may have contributed to slight discrepancies in extracted outcomes [44]. The majority of studies included in this study employed a cross-sectional design, which limits the ability to derive causal relationships and assumes a representative sample. Additionally, there are a limited number of trained cytologists and laboratory personnel to conduct quality assurance of these screening methods in LMICs, leading to variability in the quality of assessment across places and time. Finally, all systematic reviews are subject to publication bias; further, our review was limited to articles published in English. The review and meta-analysis would have benefited from the inclusion of additional languages, particularly given its global scope.

## Conclusions

Based on these findings, primary HPV testing and VIA testing demonstrated the highest diagnostic accuracy for early detection of CIN2+ in women aged 30–65 years in LMIC settings. We note that the sensitivity and specificity of VIA were high in this meta-analysis; in practice, there is substantial variability in how procedures are conducted and interpreted by clinicians. In resource-constrained settings, however, VIA may be a useful screening method given the cost-effectiveness of the test when compared to laboratory-based primary HPV testing. Given the cost and infrastructure required for the laboratory-based primary HPV testing as well as the requirement for a follow up visit, the development of low-cost point of care primary HPV testing kits that screen for 14 high-risk types of HPV and produce results in 1 to 2.5 hours holds significant promise for use in LMICs [52–54]. Recent World Health Organization guidelines also corroborate our results and conclusions of their use in LMICs; primarily, that policymakers and governments should be allocating funds and resources for clinician application of primary HPV testing among women [55]. Our review also amplifies the need for reproducible research in this topic area, examining a range of CIN thresholds, and conducting more head-to-head comparisons to better understand the performance of screening methods in low- and middle-income countries. Lastly, this review does not address demand-side considerations regarding awareness of cervical cancer screening and what efforts might be needed to improve health-seeking by women in LMICs.

## Supporting information

**S1 Table. NIH study quality assessments for observational studies (n = 11) and a randomized control trial (n = 1) (N = 12).**
(XLSX)

**S1 Fig. PRISMA 2020 checklist.**
(DOCX)

**S1 Data. Studies with data on HPV screening.**
(XLSX)

**S2 Data. Studies with data on Pap smear screening.**
(XLSX)

**S3 Data. Studies with data on VIA screening.**
(XLSX)

**S4 Data. Studies with data on VILI screening.**
(XLSX)

## Acknowledgments

We would like to express our appreciation to the global community of researchers whose work contributed to this systematic review and meta-analysis.

## Author Contributions

**Conceptualization:** Sabrina K. Smith, Oguchi Nwosu, Mohammed K. Ali.

**Data curation:** Sabrina K. Smith, Kara Suvada.

**Formal analysis:** Alex Edwards, Kara Suvada.

**Investigation:** Sabrina K. Smith, Kara Suvada, Mohammed K. Ali.

**Methodology:** Sabrina K. Smith, Michael H. Chung, Mohammed K. Ali.

**Project administration:** Sabrina K. Smith, Kara Suvada.

**Software:** Alex Edwards.

**Supervision:** Michael H. Chung, Kara Suvada, Mohammed K. Ali.

**Validation:** Mohammed K. Ali.

**Writing – original draft:** Sabrina K. Smith, Mohammed K. Ali.

**Writing – review & editing:** Sabrina K. Smith, Oguchi Nwosu, Meseret Zerihun, Kara Suvada, Mohammed K. Ali.

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
