## [Decision Letter · Decision Letter 0]

9 Nov 2022

PGPH-D-22-01579

Performance of Screening Tools for Cervical Neoplasia Among Women in Low- and Middle-income Countries: A Systematic Review and Meta-Analysis

Dear Dr. Suvada,

Thank you for submitting your manuscript to PLOS Global Public Health. After careful consideration, we feel that it has merit but does not fully meet PLOS Global Public Health’s publication criteria as it currently stands. Therefore, we invite you to submit a revised version of the manuscript that addresses the points raised during the review process.

We look forward to receiving your revised manuscript.

Kind regards,

Nebiyu Dereje, MPH, PhD

Academic Editor

Journal Requirements:

1. Please ensure you have provided an up to date search and updated the analysis as applicable.

2. Please provide separate figure files in .tif or .eps format.

4. In the online submission form, you indicated that "All data were extracted from already-published journal articles that are either openly accessible or through an institutional/individual access. Aggregate dataset is available upon reasonable request.". All PLOS journals now require all data underlying the findings described in their manuscript to be freely available to other researchers, either 1. In a public repository, 2. Within the manuscript itself, or 3. Uploaded as supplementary information.

Additional Editor Comments (if provided):

Reviewers' comments:

Reviewer's Responses to Questions

**Comments to the Author**

1. Does this manuscript meet PLOS Global Public Health’s publication criteria? Is the manuscript technically sound, and do the data support the conclusions? The manuscript must describe methodologically and ethically rigorous research with conclusions that are appropriately drawn based on the data presented.

Reviewer #1: Yes

Reviewer #2: Yes

Reviewer #3: Yes

2. Has the statistical analysis been performed appropriately and rigorously?

Reviewer #1: Yes

Reviewer #2: Yes

Reviewer #3: Yes

3. Have the authors made all data underlying the findings in their manuscript fully available (please refer to the Data Availability Statement at the start of the manuscript PDF file)?

Reviewer #1: Yes

Reviewer #2: Yes

Reviewer #3: Yes

4. Is the manuscript presented in an intelligible fashion and written in standard English?

Reviewer #1: Yes

Reviewer #2: Yes

Reviewer #3: Yes

5. Review Comments to the Author

Reviewer #1: Thank you for the opportunity to review this manuscript on the performance of screening tools for cervical neoplasia among women in low and middle income countries.

The manuscript is well written

I have the following comments:

1. The test parameters for the screening tests would look very different in a high income country. The authors did not comment on this in the discussion. It is very important as decisions on a screening test should be made on results from the test in the environment in which it will be used.

Minor comment:

1. Lines 261-263 are a repeat of lines 259-261

Reviewer #2: This study addresses an important research question and adds to the existing literature on the subject. I do have a few comments that in my opinion will strengthen this paper.

1. Whereas most readers understand the concepts of sensitivity and specificity, they may be less familiar with diagnostic odds ratio and the pooled receiver operating characteristic curve. It would be good define all of these measures (with equations if possible) early in the Methods section.

2. Some of the forest plots include just one or two studies. These results should be considered and reported, of course, but I would advise against calling it a “meta-analysis”. I suggest limiting the meta-analysis to situations where there are at least three studies.

3. Some of the results are so heterogeneous that a pooled estimate is essentially uninterpretable. In those circumstances it is more useful to explore the reasons for heterogeneity, rather than simply acknowledging its existence. This can be done qualitatively (by trying to understand the difference across studies) or quantitatively (e.g. through meta-regression analysis). Given the relative paucity of the data, a qualitative approach may be more feasible.

4. I must confess I don’t know how to read the pooled SROC curves, especially in Figures 2 and 5. They don’t look like ROC curves in a typical study. I wonder if simply reporting numeric AUC values would be better.

5. It is hard to believe that all studies were of equally high quality (as stated in the discussion). The study quality is really a relative rather than an absolute measure. Surely some studies are stronger than others. For example, only one of the 12 studies was a randomized trial. It would be good to know how its findings compare to those reported in cross sectional studies.

One minor point: Some acronyms and abbreviations (e.g. CIN2+, TP, TN, etc.) should be spelled out when they are first mentioned in the text. This is not always the case.

Reviewer #3: The topic is relevant to the target context, LMIC.

The authors have provided adequate information in all sections that allow readers to; understand where there is a problem, what is intended to be done and how the study was executed. In other words, a scientist could easy reproduce the study

The details and justifications of approaches used increase the rigor of the authors' work. Furthermore, consistency in the write up is commendable.

Few things need the attention of the authors:

Some sentences in the results (line 260-263) are repeated.

Clear implications of some of the findings e.g DOR for each test is not clear. The study could be more useful to both clinicians and policy makers. How these results are found useful to these groups is not explicit.

The recommendation, Using a point of care HPV screen test lacks clear basis. No costs analysis nor test acceptability comparisons were done in this study.

There is lack of consistency in writing the references. Some follow the standard sentence case, others have predominant capital letters. This needs to be corrected.

6. PLOS authors have the option to publish the peer review history of their article (what does this mean?). If published, this will include your full peer review and any attached files.

**Do you want your identity to be public for this peer review?** For information about this choice, including consent withdrawal, please see our Privacy Policy.

Reviewer #1: No

Reviewer #2: No

Reviewer #3: **Yes: **Davis Elias Amani

---

## [Editor Report · Decision Letter 1]

24 Jan 2023

Performance of Screening Tools for Cervical Neoplasia Among Women in Low- and Middle-income Countries: A Systematic Review and Meta-Analysis

PGPH-D-22-01579R1

Dear Suvada,

We are pleased to inform you that your manuscript 'Performance of Screening Tools for Cervical Neoplasia Among Women in Low- and Middle-income Countries: A Systematic Review and Meta-Analysis' has been provisionally accepted for publication in PLOS Global Public Health.

Best regards,

Nebiyu Dereje, MPH, PhD

Academic Editor